# Structured Populations of Critically Endangered Yellow Water Lily (*Nuphar shimadai* Hayata, Nymphaeaceae)

**DOI:** 10.3390/plants11182433

**Published:** 2022-09-19

**Authors:** Junaldo A. Mantiquilla, Hsueh-Yu Lu, Huei-Chuan Shih, Li-Ping Ju, Meng-Shin Shiao, Yu-Chung Chiang

**Affiliations:** 1Department of Biological Sciences, National Sun Yat-sen University, 70 Lienhai Road, Kaohsiung City 80424, Taiwan; 2Department of Biological Sciences and Environmental Studies, College of Science and Mathematics, University of the Philippines Mindanao, Mintal, Davao City 8022, Philippines; 3Department of Nursing, Meiho University, Pingtung 912, Taiwan; 4Fushan Research Center, Taiwan Forestry Research Institute, Yilan County 264013, Taiwan; 5Research Center, Faculty of Medicine Ramathibodi Hospital, Mahidol University, Bangkok 10400, Thailand; 6Department of Biomedical Science and Environment Biology, Kaohsiung Medical University, Kaohsiung 807, Taiwan

**Keywords:** *Nuphar shimadai* Hayata, genetic variation, population structure, simple sequence repeat (SSR), geographic isolation

## Abstract

Yellow water lily (*Nuphar shimadai* Hayata) is a critically endangered species in Taiwan. Here, we examined genetic structures of four extant populations, WP, GPa, GPb and GPn, using 39 simple sequence repeat (SSR) markers. Positive genetic correlation was observed within 50 m, beyond which no correlation was detected due to isolation by distance according to Mantel correlogram. This suggests a significant genetic structuring of the species. Besides, multilocus genotype (MLG) analysis revealed that GPa was a panmictic population and the species’ putative center of origin. Genetic exchange was observed between GPa and GPb populations, which likely resulted from their geographic proximity. Nevertheless, there was a strong asymmetric migration detected from GPa to WP, but a recent genetic barrier prevented dispersal further northward (WP). *Geneland* estimated the best number of clusters as K = 2, where WP distinctly separated from the rest of the populations. In STRUCTURE output of K = 3, a third cluster was abundant only in WP. We suggest to consider GPn and WP as separate conservation units, being far from GPa. There is indeed a need to investigate these populations; as predicted, Ne = 1.6 to 3.0 is considered low and that may put the species at risk of extinction.

## 1. Introduction

Nymphaeaceae is one of the basal groups in the ancestral lineage of the phylogeny of angiosperms [1]. This early-diverging taxon could be a possible indicator to illustrate the ecological adaptation of angiosperms under long-term geological events and climate oscillations. The *Nuphar* Sm. (Nymphaeaceae), which consisted of 11 species, originated from East Asia, and dispersed throughout the temperate region of the Northern Hemisphere, including North America, Cuba, Europe, northern Asia, and northern Africa [2].

According to nomenclature, the yellow water lily is named as *Nuphar pumila* (Timm.) DC, which is considered as a synonym to *Nuphar shimadai* Hayata [2]. However, Padgett (2007) combines *Nuphar shimadai* with *Nuphar pumila* without description and type specimen comparison. *Nuphar shimadai* was morphologically different from *N. pumila* by the stigmatic disks, which, in the former, are usually dark red, and the latter are usually yellow [3]. In addition, the distinct morphological characteristics of *N. shimadai* in Taiwan under a subtropical climate adaptation compared to *N. pumila* in temperate regions, and the geological separation and isolation, further strengthen the morphological divergence of *N. shimadai* in Taiwan from other closely related species.

A study that analyzed chloroplast genome showed that *N. pumila* (160,737 kb) has a slightly larger genome size than *N. shimadai* (160,645 kb). Using a 66 protein-coding gene dataset for the phylogenetic analysis, four *Nuphar* species (*N. advena*, *N. longifolia*, *N. pumila* and *N. shimadai*) are monophyletic and basal among all the species in the Nymphaeaceae family. *N. pumila* and *N. shimadai* are sister branches with a very high statistical support for both Maximum Likelihood and Bayesian Inference [4]. Similar results showed high statistical support for the monophyletic group of four species, and N. *pumila* and *N. shimadai* are sister taxa based on plastid phylogenomics [5].

*N. shimadai* has been categorized as a “critically endangered” species in Taiwan [6] (Figure 1). It was widely distributed in freshwater ponds and lakes around temperate regions in Taiwan, as listed in gbif.org 20 years ago [7]. Based on governmental records and reports (in Chinese), there were more than 6000 freshwater ponds and lakes in Taoyuan City a few decades ago, which mainly served as water storage for irrigation purposes. However, the number of ponds and lakes has reduced dramatically to around 200 nowadays due to several reasons: (1) the function of water storage shifted to reservoirs, (2) the conversion of freshwater ponds and lakes to agriculture and aquaculture uses, (3) the abandoned ponds were used for anthropogenic activities (being filled for building constructions). These reasons have serious impacts on the habitat and population sizes of *N. shimadai* in northern Taiwan, which resulted in the extant populations surviving in few ponds.

Several studies have proposed to evaluate the population structures of these closely related species, *N. lutea* (L.) Sm. [8], *N. japonica* DC. [9], and *N. shimadai* Hayata [10], by using cross-species molecular markers such as microsatellites. However, phylogenetic relationships of these species are unresolved using chloroplast gene *matK* and nuclear ribosomal internal transcribed spacer (ITS) regions [11].

Microsatellites (or simple sequence repeats, SSRs) are common molecular markers useful in investigating population structures. SSRs are highly variable and are classified as neutral variations which are co-dominant and widely distributed throughout the whole genome [12,13].

To be able to protect and preserve this critically endangered species, understanding its genetic structures is urgently needed. Therefore, this study aims to use SSRs to evaluate the levels of genetic diversity of Taiwanese *N. shimadai*, i.e., populations in the four ponds in Taoyuan City, Taiwan, and to understand their population structures brought about by environmental perturbations.

## 2. Results

### 2.1. Genetic Diversity of Nuphar shimadai Populations in Northern Taiwan

A total of 62 samples were collected from 4 freshwater ponds (WP, GPa, GPb and GPn) in Taoyuan City, Taiwan (Table 1). We investigated the polymorphisms of 39 SSRs across the genomes of *N. shimadai*. The diversity indices indicate that observed heterozygosity (Ho) values were generally lower than expected heterozygosity (He) in all populations, which indicated the isolation of population and habitat fragmentations of the species (Table 1). It was pronounced in the population of GPn, which was also supported with a high coefficient of inbreeding (F). In addition, this population also has the highest percentage of polymorphic loci (%P) (Table 1).

Based on the multilocus genotype (MLG) analysis where expected MLG (eMLG) estimates the number of genotypes based on the largest sample size (N = 22 in our study) [11], only the GPa population with eMLG equaled to the observed MLG (Appendix A). We further performed tests of linkage disequilibrium (LD), which tells whether the populations were subjected to other factors such as nonrandom mating among other processes. The results of rbarD showed that 3 out 4 populations were subjected to LD except the GPa population (Figure 2). Along with MLG, it further indicated that GPa is more likely to be under panmixia, i.e., a randomly mating population.

### 2.2. Population Structure of Nuphar shimadai

We further investigated the sources of variations that contribute to population differentiation using the hierarchical AMOVA and fixation indices on polymorphisms of SSRs. These tests can measure the variations at the population level such as among or within populations, and further between individuals within population or within individuals. The AMOVA results indicate that about 17% and 31% of the variations are differentiated among populations and among individuals in the populations, respectively, while more than half (52%) of these variations are within individuals (Table 2). The fixation indices can be expanded to describe population structure at different levels: inbreeding coefficient of individuals within population (F_IS_), inbreeding coefficient among populations (F_ST_), and overall inbreeding coefficient (F_IT_). We observed similar results with AMOVA tests: the fixation indices within (F_IS_) and among population (F_ST_) are 0.3736 and 0.1707, respectively, while the overall coefficient is 0.4805. Taken together, the results of AMOVA and F indices suggest that there is population structuring likely due to inbreeding.

To determine the mechanism that leads to this excess of homozygosity, we evaluate isolation-by-distance using Mantel test. Results detected significant positive correlation (r = 0.3715), which indicates that there was an isolation-by-distance, as shown in Figure 3. The patches for a kernel density estimate in the figure suggest isolation in contrast to an intact cloud of points for no isolation. In addition, a positive correlation would mean genetic similarity across space. The Mantel correlogram, an extension of Mantel test, detected genetic similarity within a short distance, about 50 m, beyond which the absence of correlation is an indication of genetic discontinuity and habitat fragmentation (Appendix A).

Among the 62 samples, three clusters were identified by principal coordinates analysis (PCoA) (Figure 4). The major cluster consisted of samples from GPa, GPb and GPn, while the remaining two clusters consisted of WP samples that comprised one larger, and the other, a smaller cluster.

Furthermore, we ask the question whether the four populations are differentiated and how they are differentiated based on the SSR polymorphisms. The estimated number of clusters (K) were determined using STRUCTURE. The results showed that two clusters (K = 2) may describe the population differentiation the best (Figure 5A,B), where samples from GPa, GPb and GPn are in the same cluster and WP in another cluster (Figure 5C). This result was supported by *Geneland* (Figure 5E,F), which showed the best estimation of 2 clusters among the 4 populations. We further set the estimated number of clusters to 3 (K = 3) to investigate whether there is differentiation within each cluster. Interestingly, a third cluster was identified in the WP population (green bars) as the subpopulation structure (Figure 5D).

Genetic barriers between the four populations were further examined by Monmonier algorithm. We found that a significant barrier was detected between WP and the three other populations (Figure 6A). Under this test, GPa and GPb show closer relationships than the other two populations. The same phenomenon was revealed by phylogenetic analysis. The phylogram showed that the southernmost populations, GPa and GPb, were in the inner nodes of the phylogenetic tree, while GPn and WP were clustered together (Figure 6B). The results indicate that GPa and GPb might be ancestral populations and the monophyletic cluster, GPn and WP, may have originated from the same ancestral population until they became genetically distinct recently.

According to the relative migration pattern, there was a statistically significant asymmetric migration from GPa to WP based on the estimated number of migrants (*Nm*) as a measure of genetic differentiation. No significant migration events were detected in other populations.

Also, the four populations of *N. shimadai* are considered as stable populations based on Wilcoxon’s test for bottleneck. Bottleneck effect results in a great reduction of genetic variations of populations due to natural disasters, among other factors. The graphical descriptor mode-shift declared all populations as “normal”, which indicates no bottleneck effect detected on the contemporary genetic data of the populations (Table 3). In addition, in line with the results from the estimated migration events mentioned previously, the Slatkin’s private allele method using *Nm* as the effective number of migrants suggests low migration events between populations (*Nm* = 0.2913) (Appendix A).

The effective population size (*Ne*) of the four populations was estimated as high as infinite and 23.1 in GPa and GPn, respectively. We hypothesized that the high level of *Ne* contributed to the current genetic diversity for GPn populations. The northernmost population, WP, obtained the least *Ne*. In general, the *Ne* estimates based on the temporal method between 0 and 1.5 generations indicated also very low with *Ne* = 1.6 to 3.0 (Table 4).

## 3. Discussion

### 3.1. Genetic Diversity Analysis Revealed GPa Population as the Center of Origin

The genus *Nuphar* remains to be resolved in terms of its phylogenetic relationships because of extreme morphological variations within and between populations [2]. This study addresses relevant information to characterize the genetic variation, particularly of the extant *N. shimadai* populations in northern Taiwan, where mention of this species in literature is scarce.

The deviations of those observed from expected heterozygosity suggest that biological or environmental processes may have shaped populations to depart from Hardy-Weinberg equilibrium in the neutral genomic SSR loci. This genetic pattern was also observed in genotypic analysis using MLG. The results in Appendix A showed that three out of four populations were subjected to LD except GPa, which was expected as a panmictic population (a population under random mating) [14]. Furthermore, the phylogenetic and migration analyses also pointed out that GPa is the putative center of origin. The other populations, showing evidence of LD, may be clonal populations. This means that the MLG count in excess of the eMLG might be ramets of a particular genet (eMLG = 11 in Appendix A), where eMLG was similar across the four populations. The high fixation index (F) of GPn (Table 1) indicated that the population may have comprised a colony of clones from these genets as *Nuphar*, generally known to form large clonal populations, and seeds predominantly dispersed by hydrochory [2].

### 3.2. Genetic Structure Detected Due to a Genetic Barrier to WP, the Northernmost Population

The phylogram showed that both GPa and GPb are ancestral (Figure 6B). The branch lengths of the phylogram are proportional to the inferred evolutionary change. However, it is unrooted, meaning that the root has not been hypothesized [15]. Microsatellites are known to serve as accurate molecular clocks for coalescent times of at least 2 million years, and an unbiased estimate for genetic differentiation to infer population history. Unfortunately, the SSR markers selected in this study only resolved as far as 150,000 years back in time (Figure 6B). Meantime, GPn and WP are monophyletic, i.e., they belong to the same clade until a genetic barrier caused the separation of these populations that resulted in this dichotomy (Figure 6A).

Particularly GPb, which is significant to all heterozygosity excess, might have diminished bottleneck footprints due to recent genetic exchange with GPa based on proximity (Appendix A). GPa may have genetic connectivity with GPn, but not strong enough to be detected as statistically significant by relative migration patterns, as gene flow was also low (*Nm* = 0.291) (Appendix A). None was also detected as first (F_0_)-generation immigrants between GPa and GPn populations (Appendix A). In a simulation using the genetic software GENECLASS2, F_0_ immigrants are considered “misassigned” based on their genotypes, because they were born somewhere else other than their respective home populations [16]. A similar genetic pattern was observed on *N. submersa*, a critically endangered freshwater macrophyte indigenous to central Japan [17]. The average level of gene flow (*Nm* = 0.122) was much lower compared to the extant *N. shimadai* populations in Taiwan from the current study.

Surprisingly, there was a strong relative migration from GPa as a source to WP. It is likely that this strong directional gene flow between these two populations occurred in the distant past that influenced the composition of individuals in the populations recently. The footprints can either enhance or obscure the signs of recent migration in the genetic data [18]. Based on the Monmonier plot (Figure 6A), the detected genetic barrier appeared to contribute to population fragmentation, resulting in population differentiation. When the number of estimated clusters was set to three (K = 3), an extra cluster (green) was observed in WP predominantly.

Among the northern Taiwan populations, WP has the lowest estimate of contemporary effective population size (*Ne* = 2.3) (Table 4). If this trend continues, it will be on top of the list at risk of extinction. The GPa population has *Ne* = infinity, i.e., the results can be explained by sampling error without necessarily attributing to drift [19]. Simply, the population is large enough. Nevertheless, these populations are generally slow to bounce back with a range of *Ne* between 1.6 and 3.0, as projected in a simulation within 1.5 generations.

As STRUCTURE analysis indicated that either two (K = 2) or three (K = 3) estimated clusters can well differentiate clusters involving 4 extant populations within 1 km in still water or ponds, we thus suggest to consider populations far apart as separate management units. There is a reason for GPa and GPb to be considered as one, not only physically, but by proximity based on genetic exchange. As Mantel correlogram indicated that correlation was only detected within 50 m, GPn and WP can be treated as separate units because of the distance from GPa, as a putative center of diversity.

## 4. Materials and Methods

### 4.1. Sampling and DNA Extraction

A survey was conducted of the previously identified sites where populations of *Nuphar shimadai* existed had gone extinct due to urbanization, as we mentioned in the introduction, except for the four populations found in the protected ponds in Taoyuan City, northern Taiwan. There were 62 samples collected at 20 m apart to avoid clones from the same individuals (Table 5). The plant tissues were kept dry using silica gel [20]. The genomic DNA samples were extracted using a Genomic DNA Extraction Kit (RBC Bioscience, Taipei, Taiwan).

### 4.2. SSR Selection and Amplification

A microsatellite-enriched library was conducted through magnetic bead procedure [21,22]. Out of 72 SSR markers, 39 were successfully amplified to evaluate genetic variability of the populations. The development of these SSR markers was already reported by [10] under the following PCR conditions during optimization by temperature gradient: 94 °C for 2 min; 35 cycles of 94 °C for 45 sec; 50 to 60 °C for 45 sec; and 72 °C for 50 sec; followed by the final extension set at 72 °C for 7 min. The amplification was conducted using the following 20 µL mixture: 5 ng of genomic DNA, 4 µL of 5× buffer, 0.2 µM of dNTP mix, 2 mM of MgCl_2_, 0.5 units of GoTaq MDx Hot Start Polymerase (Promega Corporation), 0.2 mM of both primers, and sterile ddH_2_O. PCR amplification was performed using a Labnet Multigene 96-well Gradient Thermal Cycler (Labnet, Edison, NJ, USA). The amplicons were checked with 1% agarose gel electrophoresis (please see Appendix A for PCR amplicons of the selected 10 primers, and for more information of published primer sequences, please refer to reference [10]). The target amplicons were visualized using a digital camera (Top BIO Co., Taipei, Taiwan) after being electrophoresed in a 10% polyacrylamide gel. Amplicon sizes were scored as co-dominant using Quantity One^®^ software (Bio-Rad) to create a data matrix manually.

### 4.3. Molecular Data Analysis

Data analyses and visualizations were performed using R Stats Package [23] unless otherwise indicated. The genetic diversity analyses were conducted using GeneAlEx [24] and the multilocus genotype (MLG) analysis by *poppr* in R, where departure from linkage disequilibrium indicates a panmictic population [25].

The proportion of each hierarchical variation was evaluated using the Analysis of Molecular Variance (AMOVA) by Arlequin [26], while the genetic relationship of the individuals was obtained through principal coordinate analysis (PCoA) in kernel density analysis to evaluate spatial clustering of samples using *adegenet* in R [27]. Additional testing for spatial patterns was conducted to detect genetic boundaries among georeferenced genotypes using the Monmonier algorithm [28].

Clustering analysis was also performed by using STRUCTURE [29] in 10 runs for each subpopulation from K = 1 to K = 4 with a burn-in period of 2 × 10^5^ generations after 2 × 10^6^ MCMC replicates per run. The results were illustrated by STRUCTURE HARVESTER to determine the best K according to the Evanno method [30]. Like STRUCTURE, *Geneland* is another Bayesian-model clustering analysis, but it runs georeferenced genotype data [31]. The most probable number of K was based on the highest average posterior probability from 10 runs in 1 × 10^5^ MCMC iterations under spatial and nonspatial models in uncorrelated assumption of allele frequency. The output of the Poisson–Voronoi tessellation was determined based on 200 × 200 pixels with a burn-in of 200.

The directional migration pattern was simulated by *diveRsity* implemented in R based on relative migration network using *Nm* as a measure of genetic differentiation [18]. Moreover, the first-generation migrant detection was conducted by GENECLASS2 based on computation criterion from allele frequency and a Monte Carlo resampling method of 1000 individuals with *p* = 0.01 [16]. On the other hand, the *Nm* as an estimate of the effective number of migrants per generation based on Slatkin’s private allele method was estimated using GENEPOP as an R Package [32]. The phylogenetic tree as presented by phylogram was based on the Neighbor-Joining method according to Nei’s distance implemented by *adegenet* in R to visualize the population clustering in northern Taiwan [27].

The bottleneck test examining populations that experienced a severe reduction in effective population size (*Ne*) was performed by the software BOTTLENECK. In this test, the heterozygosity excess was compared to expected equilibrium heterozygosity (H_e_ > H_eq_) under mutation-drift equilibrium. It was determined statistically by Wilcoxon’s signed rank test [33]. The *Ne*, as a contemporary effective population size, was estimated by NeEstimator v 2.1 based on linkage disequilibrium under random mating with a critical value (P_crit_ ≥ 0.02) on allele frequency for rare alleles. Both the single-sample method and the two-sample or temporal method under a chi-squared parametric 95% confidence interval were performed for the *Ne* estimate [34].

## Figures and Tables

**Figure 1 plants-11-02433-f001:**
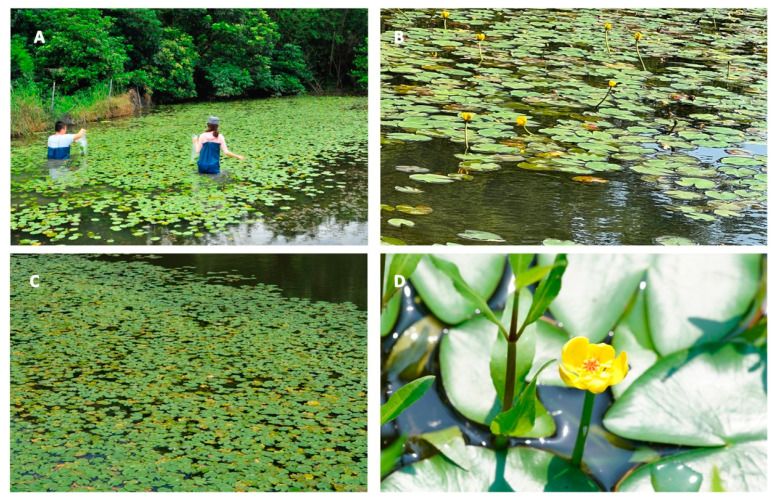
The four *N. shimadai* populations in freshwater ponds found in different areas of Taoyuan City, northern Taiwan: (**A**) WP, (**B**) GPa, (**C**) GPb and (**D**) GPn.

**Figure 2 plants-11-02433-f002:**
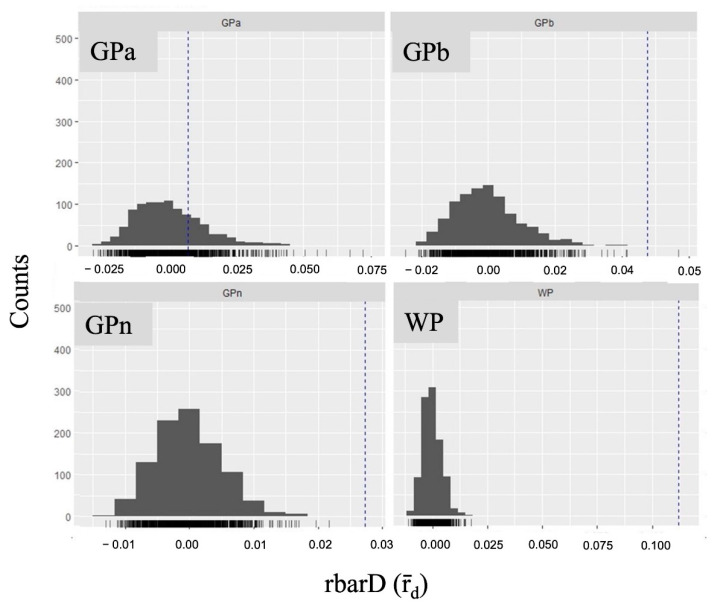
The standardized index of association, rbarD (r_d_), of four *N. shimadai* populations in northern Taiwan. Vertical dashed lines determine statistical significance (*p* = 0.05) for linkage disequilibrium. Only the GPa population is under panmixia as it is not significant to LD. The other three populations showed evidence of LD (*p* < 0.05).

**Figure 3 plants-11-02433-f003:**
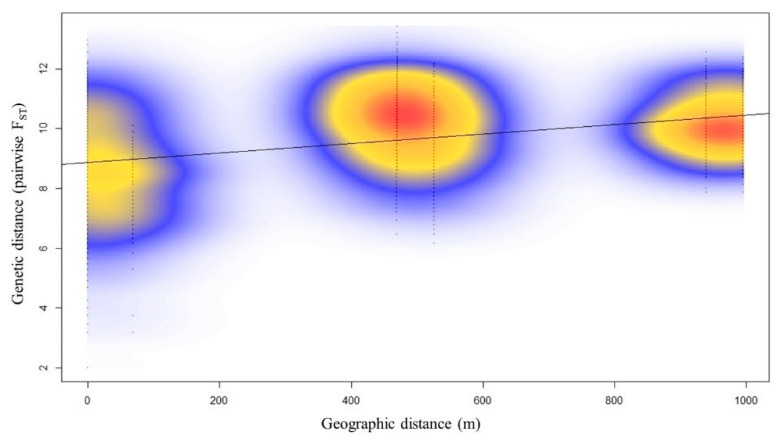
Isolation-by-distance scatterplot using kernel density estimate of *N. shimadai* between the matrix of genetic distance (pairwise F_ST_) and geographic distance (meters) in northern Taiwan. The genetic distance and geographic distance show a significant positive correlation (r = 0.3715).

**Figure 4 plants-11-02433-f004:**
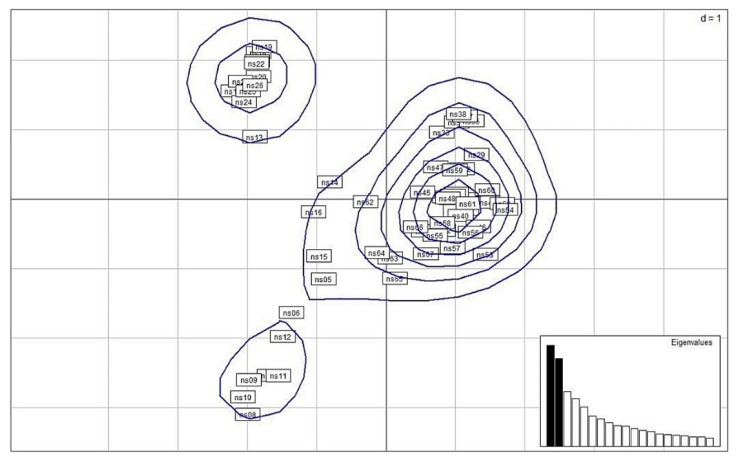
The principal coordinates analysis (PCoA) of *N. shimadai* samples from 4 populations in northern Taiwan based on the kernel density estimate of individuals using axes (PC) 1 and 2.

**Figure 5 plants-11-02433-f005:**
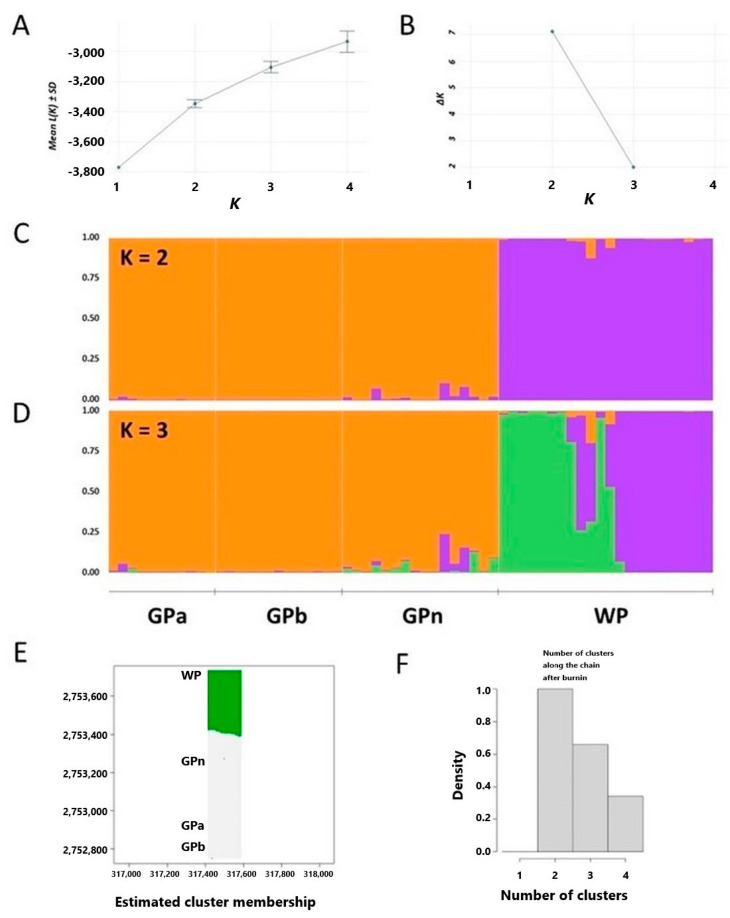
The estimated number of clusters (K) of the four *N. shimadai* populations in northern Taiwan based on the results of the Evanno method in STRUCTURE output. (**A**) Mean L/K (estimated probability value for each K value), (**B**) inferred cluster based on delta K (ΔK), genetic clusters based on K = 2 (**C**) and K = 3 (**D**), and (**E**) *Geneland* analysis for georeferenced genotypes where (**F**) K = 2 has the highest probability based on the convergence of MCMC iterations.

**Figure 6 plants-11-02433-f006:**
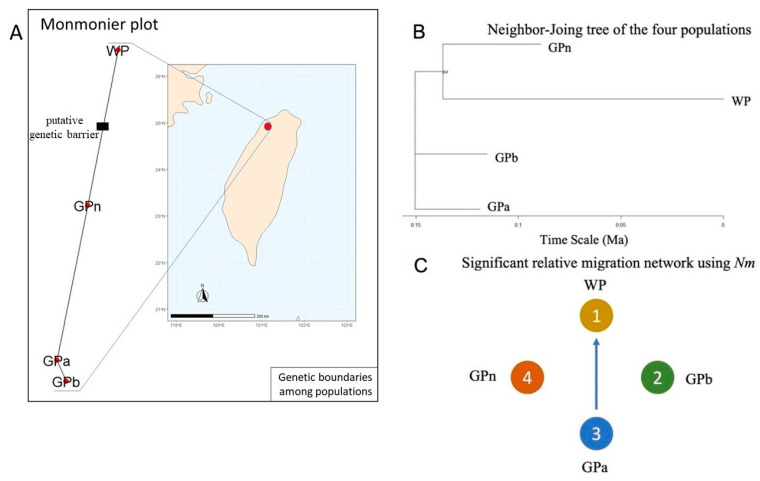
The genetic relationships of *N. shimadai* populations in northern Taiwan: (**A**) the genetic barrier (black square) in the Monmonier plot was detected between the WP population and the others, (**B**) the phylogram according to time scale (in Ma) by the Neighbor-Joining method based on 39 microsatellite loci (bootstrap = 1000), and (**C**) a significant (*p* < 0.05) migration was detected from the GPa to the WP population based on *Nm* indicated by the blue arrow. The *divMigrate* filter threshold function at 0.5 for asymmetric values and bootstrap = 1000 were applied.

**Table 1 plants-11-02433-t001:** Indices of genetic diversity from 4 populations of *N. shimadai* in northern Taiwan using 39 genomic SSR loci.

Populations	Diversity Indices								
	N	N_a_	N_e_	I	H_o_	H_e_	uH_e_	F	%P
WP	22	2.795	1.930	0.687	0.248	0.402	0.411	0.444	87.18
GPb	13	1.974	1.600	0.431	0.221	0.266	0.276	0.344	61.54
GPa	11	2.077	1.554	0.444	0.228	0.271	0.284	0.412	71.79
GPn	16	2.744	1.822	0.652	0.199	0.383	0.395	0.612	94.87
Mean		2.390	1.727	0.554	0.224	0.331	0.342	0.453	78.85
SE		0.100	0.059	0.033	0.028	0.019	0.019	0.055	7.50

N = sample size; N_a_ = number of different alleles; N_e_ = number of effective alleles; I = Shannon’s Information Index; H_o_ = observed heterozygosity; H_e_ = expected heterozygosity; uH_e_ = unbiased expected heterozygosity; F = population’s fixation index; %P = percent polymorphic loci; SE = standard error.

**Table 2 plants-11-02433-t002:** Analysis of molecular variance (AMOVA) of *N. shimadai* populations in northern Taiwan using 39 SSR loci.

Source of Variation	Sum of Squares	Variance Components	Percentage Variation	*p*-Value
Among populations	160.56	1.4491	17.0657	0.0000 *
Among individuals Within populations	560.64	2.6309	30.9835	0.0000 *
Within individuals	273.50	4.4113	51.9508	0.0000 *
Total	994.70	8.4913		

* Significant at *p* < 0.05.

**Table 3 plants-11-02433-t003:** The Wilcoxon’s tests for bottleneck of *N. shimadai* populations in northern Taiwan across 39 SSR loci ^z^. The numbers are *p*-values under different models.

Population	IAM 1-Tailed	IAM 2-Tailed	TPM 1-Tailed	TPM 2-Tailed	SMM 1-Tailed	SMM 2-Tailed	Mode-Shift
WP	**0.0000**	**0.0000**	**0.0013**	**0.0027**	0.1140	0.2280	normal
GPb	**0.0006**	**0.0012**	**0.0044**	**0.0087**	**0.0302**	0.0604	normal
GPa	**0.0226**	**0.0451**	0.0929	0.1858	0.5401	0.9375	normal
GPn	**0.0055**	**0.0110**	0.1473	0.2947	0.6834	0.6438	normal

^z^ Boldface indicates significance (*p* < 0.05) in tests either 1-tailed (for heterozygosity excess) or 2-tailed (for heterozygosity excess or deficit) under infinite allele model (IAM), two-phase mutation (TPM) or stepwise mutation model (SMM). Mode-shift is a graphical descriptor of the allele frequency distribution which distinguishes bottlenecked over stable populations.

**Table 4 plants-11-02433-t004:** The effective population size (*Ne*) of *N. shimadai* in northern Taiwan estimated from linkage disequilibrium under random mating with allele frequency ≥ 0.02 ^a^.

Populations	Single-Sample		Two-Sample	
	*Ne*	95% CI	*Ne*	95% CI
WP	2.3	2.1–2.6		
GPb	11.7	6.1–26.9	1.6	1.1–2.1
GPa	Infinite	Infinite		
GPn	23.1	15.4–40.0	3.0	2.1–4.0

^a^ CI—confidence interval; infinity means a negative estimate of *Ne*, which can be explained by sampling error; the two-sample (temporal) method under Plan II (juvenile individuals sampled without replacement) utilized two populations sampled between 0 and 1.5 generations apart as default according to Jorde and Ryman (2007).

**Table 5 plants-11-02433-t005:** Sampling sites of the four *Nuphar shimadai* populations in northern Taiwan.

Populations	Sampling Sites	Longitude	Latitude	*N*
WP	Taoyuan, Taiwan	121°11′39″ E	24°53′16″ N	22
GPb	Taoyuan, Taiwan	121°11′34″ E	24°52′44″ N	13
GPa	Taoyuan, Taiwan	121°11′33″ E	24°52′46″ N	11
GPn	Taoyuan, Taiwan	121°11′36″ E	24°53′01″ N	16

## Data Availability

The datasets generated for this study are available on request to the corresponding author.

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
