# Peer review of "Structured Populations of Critically Endangered Yellow Water Lily (*Nuphar shimadai* Hayata, Nymphaeaceae)"

_plants, 2022, doi:10.3390/plants11182433_

Round 1
Reviewer 1 Report
Authors should revise some points:
1. Lines 43-51: Authors should add references for each sentence.
2. Line 329: Authors should provide the figure of the amplicons which were checked with 1% agarose gel electrophoresis.
3. Authors wrote the manuscript carefully. Authors should provide the figure flower of four populations of Nuphar shimadai.
Author Response
Thank you very much! I revised our manuscript according to your valuable comments.
Reviewer 1
- Lines 43-51: Authors should add references for each sentence.
Reply: Thank you for the suggestion. The changes of irrigation cannel and pounds were from the investigation of local government and reported as the historical record in Chinese. Therefore, we indicated as “Based on the governmental record and report (in Chinese),” in Line 63.
- Line 329: Authors should provide the figure of the amplicons which were checked with 1% agarose gel electrophoresis.
Reply: Thank you for the suggestion.We provided the electrophoresis gel photos of PCR amplicons of 10 selected primer pairs in Supplementary document 1 due to file size limitation. We also added the information in Line 451-452. as “Please see Supplementary Documents 1 for PCR amplicons of selected 10 primers and for more information of published primer sequences, please refer reference [7]”
- Authors wrote the manuscript carefully. Authors should provide the figure flower of four populations of Nuphar shimadai.
Reply: Thank you for the suggestion.We added new picture of yellow water lily as Figure 1 (Line 86 and 108).

Reviewer 2 Report
line
4 Junaldo A. Mantiquilla 1,2+, > Junaldo A. Mantiquilla 1,2,‡,
4 Hsueh-Yu Lu 1+ > Hsueh-Yu Lu 1,‡
11 Taiwan; > Taiwan
15 * Correspondence: > decrease font size
15-16 * Correspondence: Y.C.C.; yuchung@mail.nsysu.edu.tw; 886-7-5252000 ext 3625; M.S.S.; msshiao@gmail.com; 66-2-2011000 ext 1910; L.P.J; lpju57@gmail.com; 886-3-9228900 ext 113
>
* Correspondence: yuchung@mail.nsysu.edu.tw; Tel.: 886-7-5252000 ext 3625 (Y.C.C.); msshiao@gmail.com; Tel.: 66-2-2011000 ext 1910 (M.S.S.); lpju57@gmail.com; Tel.; 886-3-9228900 ext 113 (L.P.J.)
17 † These > ‡ These
Table 1, some mean values have to be corrected:
2.397 > 2.398
1.726 > 1.727
0.330 > 0.331
0.468 > 0.453
82,90 insert empty row after the line
83-85 this sentence is not clear and you have not Table S1
119 (51%) > (52%)
137,230 you have not Table S2
178 (Figure 4E – G) > you have not Figure 4G
196 increase font size of the chart axes of the Figure 4A-F for better readability
217 add large and bold letter “A” on the chart (up left)
258,263 Table S1 > you have not Table S1
280,283 Table S3 > you have not Table S3
282 Table S4 > you have not Table S4
323,324 72 °C > 72°C
326 MgCL2 > MgCl2
327 ddH2O > ddH2O
369-370 missing link for supporting information
371-376
Author Contributions: Conceptualization, H.Y.L., L.P.J., Y.C.C.; methodology, H.Y.L., Y.C.C.; software, J.A.M; validation, J.A.M., Y.C.C.; formal analysis, J.A.M.; investigation, J.A.M., H.Y.L., L.P.J., H.C.S, Y.C.C.; resources, H.Y.L., L.P.J., Y.C.C.; data curation, J.A.M., Y.C.C.; writing-original draft preparation, J.A.M., H.Y.L, M.S.S.; Y.C.C., writing-review and editing, J.A.M., Y.C.C., M.S.S.; visualization, J.A.M.; supervision, Y.C.C.; project administration, Y.C.C.; funding acquisition, L.P.J., Y.C.C.
>
Author Contributions: Conceptualization, H.Y.L., L.P.J. and Y.C.C.; methodology, H.Y.L. and Y.C.C.; software, J.A.M.; validation, J.A.M. and Y.C.C.; formal analysis, J.A.M.; investigation, J.A.M., H.Y.L., L.P.J., H.C.S. and Y.C.C.; resources, H.Y.L., L.P.J. and Y.C.C.; data curation, J.A.M. and Y.C.C.; writing-original draft preparation, J.A.M., H.Y.L, M.S.S. and Y.C.C.; writing-review and editing, J.A.M., Y.C.C. and M.S.S.; visualization, J.A.M.; supervision, Y.C.C.; project administration, Y.C.C.; funding acquisition, L.P.J. and Y.C.C.
386-445 add doi for references where is available and missing
434 link is not functional
Author Response
Thank you very much! I revised our manuscript according to your valuable comments.
Reviewer 2
Reply: Thank you for the detailed correction. We corrected all the typos and indicated them in the new line number accordingly.
|
(1) L4 Junaldo A. Mantiquilla 1,2+, > Junaldo A. Mantiquilla 1,2,‡, (2) 4 Hsueh-Yu Lu 1+ > Hsueh-Yu Lu 1,‡ (3) Taiwan; > Taiwan (4) * Correspondence: > decrease font size (5) 15-16 * Correspondence: Y.C.C.; yuchung@mail.nsysu.edu.tw; 886-7-5252000 ext 3625; M.S.S.; msshiao@gmail.com; 66-2-2011000 ext 1910; L.P.J; lpju57@gmail.com; 886-3-9228900 ext 113 (6) † These > ‡ These
|
Thank you for the suggestion. We have corrected all these accordingly and label them by tracking function. |
|
Table 1, some mean values have to be corrected:
2.397 > 2.398
1.726 > 1.727
0.330 > 0.331
0.468 > 0.453
|
Thank you for the suggestion. We have corrected all these accordingly and label them by tracking function. |
|
82,90 insert empty row after the line |
We have corrected all these accordingly |
|
83-85 this sentence is not clear and you have not Table S1 |
Thank you for the suggestion. We have Revised sentence: L122 – L125 and added Table S1 in the end of the article “Based on the multilocus genotype (MLG) analysis where expected MLG (eMLG) estimates the number of genotypes based on the largest sample size (N = 22 in our study) [11], only GPa population with eMLG equaled to the observed MLG (Table S1).” |
|
119 (51%) > (52%)
|
We have corrected it in L171 |
|
137,230 you have not Table S2
|
We have added all the supplementary Tables in the end of the article. |
|
178 (Figure 4E – G) > you have not Figure 4G
|
Thank you for the suggestion. We have changed accordingly. In addition, due to adding one more figure (Figure 1) requested by Reviewer 1, Figure 4 is now Figure 5 in the text. Please refer to L303. |
|
196 increase font size of the chart axes of the Figure 4A-F for better readability
|
We have changed accordingly |
|
217 add large and bold letter “A” on the chart (up left)
|
We have added A in figure 6 (originally Figure 5) |
|
258,263 Table S1 > you have not Table S1
|
We have added all the supplementary Tables in the end of the article. |
|
280,283 Table S3 > you have not Table S3
|
We have added all the supplementary Tables in the end of the article. |
|
282 Table S4 > you have not Table S4
|
We have added all the supplementary Tables in the end of the article. |
|
323,324 72 °C > 72°C
|
Revised text in Line 527 |
|
326 MgCL2 > MgCl2
|
Revised text: L529 |
|
327 ddH2O > ddH2O
|
Revised text: L530 |
|
369-370 missing link for supporting information
|
This will be provided by MDPI editorial board later on after acceptance of the manuscript |
|
L371 – 376 Revise Author Contributions: Conceptualization, H.Y.L., L.P.J., Y.C.C.; methodology, H.Y.L., Y.C.C.; software, J.A.M; validation, J.A.M., Y.C.C.; formal analysis, J.A.M.; investigation, J.A.M., H.Y.L., L.P.J., H.C.S, Y.C.C.; resources, H.Y.L., L.P.J., Y.C.C.; data curation, J.A.M., Y.C.C.; writing-original draft preparation, J.A.M., H.Y.L, M.S.S.; Y.C.C., writing-review and editing, J.A.M., Y.C.C., M.S.S.; visualization, J.A.M.; supervision, Y.C.C.; project administration, Y.C.C.; funding acquisition, L.P.J., Y.C.C. à Author Contributions: Conceptualization, H.Y.L., L.P.J. and Y.C.C.; methodology, H.Y.L. and Y.C.C.; software, J.A.M.; validation, J.A.M. and Y.C.C.; formal analysis, J.A.M.; investigation, J.A.M., H.Y.L., L.P.J., H.C.S. and Y.C.C.; resources, H.Y.L., L.P.J. and Y.C.C.; data curation, J.A.M. and Y.C.C.; writing-original draft preparation, J.A.M., H.Y.L, M.S.S. and Y.C.C.; writing-review and editing, J.A.M., Y.C.C. and M.S.S.; visualization, J.A.M.; supervision, Y.C.C.; project administration, Y.C.C.; funding acquisition, L.P.J. and Y.C.C.
|
Thank you for the suggestion. We have modified accordingly: Conceptualization, H.Y.L., L.P.J. and Y.C.C.; methodology, H.Y.L. and Y.C.C.; software, J.A.M; validation, J.A.M. and Y.C.C.; formal analysis, J.A.M.; investigation, J.A.M., H.Y.L., L.P.J., H.C.S and Y.C.C.; resources, H.Y.L., L.P.J. and Y.C.C.; data curation, J.A.M. and Y.C.C.; writing-original draft preparation, J.A.M., H.Y.L, M.S.S. and Y.C.C., writing-review and editing, J.A.M., Y.C.C. and M.S.S.; visualization, J.A.M.; supervision, Y.C.C.; project administration, Y.C.C.; funding acquisition, L.P.J., Y.C.C. All authors have read and agreed to the published version of the manuscript.
|
|
386-445 add doi for references where is available and missing
|
We have already added Doi |
|
434 link is not functional
|
We have provided a new link |
|
|
|

Reviewer 3 Report
In the presented manuscript, the authors studied the genetic variability and genetic structure of four local populations of Nufar shimadai from northern Taiwan using 39 SSR markers. The authors used many methods of population genetics. All methods were applied correctly, and the results obtained were discussed in connection with the recent history of the studied populations and the population-genetic processes occurring in them. These results made it possible to identify the most threatened populations that are at risk of extinction. The work is solid and makes a good impression.
There are only minor items to comment:
1. I would recommend adding a short paragraph with a taxonomic note about the species Nuphar shimadai, because in Padgett's monograph (2007) and in the taxonomic databases ipni.org and POWO this name is not accepted and is considered as a synonym of Nuphar pumila or Nuphar pumila ssp. pumila. The authors should explain why they consider N. shimadai to be an independent species and what this decision is based on (morphology, molecular data, etc.).
2. Page 2, lines 53-55: -“Several studies have proposed using microsatellite markers to study population structures of these closely related species, N. lutea (L.) Sm. [4], N. japonica DC. [5], and N. shimadai Hayata [6] using cross-species molecular markers such as microsatellites”--- Please, rephrase. "using microsatellite markers" and "using molecular markers such as microsatellites" used two times.
3. I would recommend adding a short paragraph with a summary of the phylogenetic relationships of Nuphar shimadai and related species. There are many papers studying the phylogenetics and phylogenomics of Nuphar and Nymphaeaceae, for example:
Biswal, D.K., Debnath, M., Kumar, S. et al. Phylogenetic reconstruction in the Order Nymphaeales: ITS2 secondary structure analysis and in silico testing of maturase k (matK) as a potential marker for DNA bar coding. BMC Bioinformatics 13 (Suppl 17), S26 (2012). https://doi.org/10.1186/1471-2105-13-S17-S26
Dingxuan He, Andrew W. Gichira, Zhizhong Li et al., Int. J. Mol. Sci. 2018, 19, 3780; doi:10.3390/ijms19123780
Cornelia Löhne, Thomas Borsch and John H. Wiersema. Phylogenetic analysis of Nymphaeales using fast-evolving and noncoding chloroplast markers. Botanical Journal of the Linnean Society, 2007, 154, 141–163. https://doi.org/10.1111/j.1095-8339.2007.00659.x
Michael Gruenstaeudl, Lars Nauheimer, Thomas Borsch. Plastid genome structure and phylogenomics of Nymphaeales: conserved gene order and new insights into relationships. Plant Syst Evol (2017) 303:1251–1270. DOI 10.1007/s00606-017-1436-5
Gruenstaeudl M. Why the monophyly of Nymphaeaceae currently remains indeterminate: an assessment based on gene‑wise plastid phylogenomics. Plant Systematics and Evolution (2019) 305:827–836. https://doi.org/10.1007/s00606-019-01610-5
….. and others
4. All supplementary materials (Tables S1, S2, S3, S4 etc.) are unavailable.
5. The references to Figures 2 and 3 seem to be incorrect. They are mixed up.
6. A simulation using the genetic software GENECLASS2 is described in the Results, but absent from Material and Methods.
7. According to gbif.org, the distribution area of Nuphar pumila and Nuphar shimadai in Taiwan is larger, than was studied in the present paper. Why other populations of Nuphar were not included in the study?
8. What was the minimal distance between any two individuals sampled for the study? How long can the rhizome be? – Some of these questions are briefly discussed in the discussion, but the sampling strategy should be described in the Materials and Methods.
Comments are also added to pdf-file.

Author Response
Thank you very much! We revised our manuscript according to your valuable comments.
Reviewer 3
- I would recommend adding a short paragraph with a taxonomic note about the species Nuphar shimadai, because in Padgett's monograph (2007) and in the taxonomic databases ipni.org and POWO this name is not accepted and is considered as a synonym of Nuphar pumila or Nuphar pumila ssp. pumila. The authors should explain why they consider N. shimadai to be an independent species and what this decision is based on (morphology, molecular data, etc.).
Reply: Thank you for the comment. We have added two more paragraphs addressing the phylogeny of yellow water lily in Taiwan. Please see Line 44 and onwards.
According to nomenclature, the yellow water lily is named as Nuphar pumila (Timm.) DC which is considered as synonym to Nuphar shimadai Hayata [2]. However, Padgett (2007) combines Nuphar shimadai to Nuphar pumila without description and type specimen comparison. Nuphar shimadai was morphologically different from N. pumila by the stigmatic disks, which the former usually are dark red and the latter usually are yellow [3]. In addition, the distinct morphological characteristics of N. shimadai in Taiwan under a subtropical climate adaptation compared to N. pumila in temperate regions, and the geological separation and isolation further strengthen the morphological divergence of N. shimadai in Taiwan from other closely related species.
- Page 2, lines 53-55: -“Several studies have proposed using microsatellite markers to study population structures of these closely related species, N. lutea (L.) Sm. [4], N. japonica DC. [5], and N. shimadai Hayata [6] using cross-species molecular markers such as microsatellites”--- Please, rephrase. "using microsatellite markers" and "using molecular markers such as microsatellites" used two times.
Reply: Thank you for the nice comment. We have modified the sentence as follows: “Several studies have proposed to evaluate the population structures of these closely related species, N. lutea (L.) Sm. [5], N. japonica DC. [6], and N. shimadai Hayata [7] using cross-species molecular markers such as microsatellites.” In Line 72-73.
- I would recommend adding a short paragraph with a summary of the phylogenetic relationships of Nuphar shimadai and related species. There are many papers studying the phylogenetics and phylogenomics of Nuphar and Nymphaeaceae, for example:
Reply: We have combined together with comment 1 to address the phylogeny of yellow water lily in Taiwan. Please see Line 53-60
A study analyzed chloroplast genome showed that N. pumila (160,737 kb) has slightly larger genome size than N. shimadai (160,645 kb). Using a 66 protein-coding gene dataset for the phylogenetic analysis, four Nuphar species (N. advena, N. longifolia, N. pumila and N. shimadai) are monophyletic and basal among all the species in the family Nymphaeaceae. N. pumila and N. shimadai are sister branches with a very high statistical support for both Maximum Likelihood and Bayesian Inference [4]. Similar results showed high statistical support for the monophyletic group of four species and N. pumila and N. shimadai are sister taxa based on plastid phylogenomics [5].
Biswal, D.K., Debnath, M., Kumar, S. et al. Phylogenetic reconstruction in the Order Nymphaeales: ITS2 secondary structure analysis and in silico testing of maturase k (matK) as a potential marker for DNA bar coding. BMC Bioinformatics 13 (Suppl 17), S26 (2012). https://doi.org/10.1186/1471-2105-13-S17-S26
Dingxuan He, Andrew W. Gichira, Zhizhong Li et al., Int. J. Mol. Sci. 2018, 19, 3780; doi:10.3390/ijms19123780
Cornelia Löhne, Thomas Borsch and John H. Wiersema. Phylogenetic analysis of Nymphaeales using fast-evolving and noncoding chloroplast markers. Botanical Journal of the Linnean Society, 2007, 154, 141–163. https://doi.org/10.1111/j.1095-8339.2007.00659.x
Michael Gruenstaeudl, Lars Nauheimer, Thomas Borsch. Plastid genome structure and phylogenomics of Nymphaeales: conserved gene order and new insights into relationships. Plant Syst Evol (2017) 303:1251–1270. DOI 10.1007/s00606-017-1436-5
Gruenstaeudl M. Why the monophyly of Nymphaeaceae currently remains indeterminate: an assessment based on gene‑wise plastid phylogenomics. Plant Systematics and Evolution (2019) 305:827–836. https://doi.org/10.1007/s00606-019-01610-5….. and others
- All supplementary materials (Tables S1, S2, S3, S4 etc.) are unavailable.
Reply: we apologize for not being aware of the missing supplementary tables. They are added properly in the end of the article.
- The references to Figures 2 and 3 seem to be incorrect. They are mixed up.
Reply: we have modified accordingly
- A simulation using the genetic software GENECLASS2 is described in the Results, but absent from Material and Methods.
Reply: We have added the description in Materials and Methods: Moreover, the first-generation migrant detection was conducted by GENECLASS2 based on computation criterion from allele frequency and a Monte Carlo resampling method of 1000 individuals with P = 0.01 [13] in Line 478-480.
- According to gbif.org, the distribution area of Nuphar pumila and Nuphar shimadai in Taiwan is larger, than was studied in the present paper. Why other populations of Nuphar were not included in the study?
Reply: Thank you for bringing up the issue. The record of Nuphar in Taiwan in gbif were dated before 2002. The changes of irrigation system and pounds changed dramatically in the past 20 years due to human activities. That’s why the species has been listed as critical endangered species. We have make this point clearer by adding one sentence in Introduction (Line 61-63): “It was widely distributed in freshwater ponds and lakes around temperate regions in Taiwan as listed in gbif.org 20 years ago (https://www.gbif.org/occurrence/search?taxon_key=4940220)”
- What was the minimal distance between any two individuals sampled for the study? How long can the rhizome be? – Some of these questions are briefly discussed in the discussion, but the sampling strategy should be described in the Materials and Methods.
Reply: Thank you for the suggestion. We have added more sampling strategy in the beginning of the Materials and Methods (Line 430 to 434): A survey of the previously identified sites where populations of Nuphar shimadai existed had gone extinct due urbanization as we mentioned in introduction, except for the four populations found in the protected ponds in Taoyuan City, northern Taiwan. There were 62 samples collected at 20 m apart to avoid clones from the same individuals (Table 5).
